# *Roseburia intestinalis* Modulates PYY Expression in a New a Multicellular Model including Enteroendocrine Cells

**DOI:** 10.3390/microorganisms10112263

**Published:** 2022-11-15

**Authors:** Thomas Gautier, Nelly Fahet, Zohreh Tamanai-Shacoori, Nolwenn Oliviero, Marielle Blot, Aurélie Sauvager, Agnes Burel, Sandrine David-Le Gall, Sophie Tomasi, Sophie Blat, Latifa Bousarghin

**Affiliations:** 1Institut NUMECAN, INSERM, Univ Rennes, INRAE, F-35000 Rennes, France; 2ISCR (Institut des Sciences Chimiques de Rennes)-UMR CNRS 6226, Univ Rennes, CNRS, F-35000 Rennes, France; 3Plateforme Microscopie Electronique MRic/ISFR Biosit/Campus Santé, Univ Rennes, F-35000 Rennes, France

**Keywords:** hormone-producing cells, quadricellular model, gut commensal bacteria, butyrate, proglucagon, peptide tyrosine tyrosine

## Abstract

The gut microbiota contributes to human health and disease; however, the mechanisms by which commensal bacteria interact with the host are still unclear. To date, a number of in vitro systems have been designed to investigate the host–microbe interactions. In most of the intestinal models, the enteroendocrine cells, considered as a potential link between gut bacteria and several human diseases, were missing. In the present study, we have generated a new model by adding enteroendocrine cells (ECC) of L-type (NCI-H716) to the one that we have previously described including enterocytes, mucus, and M cells. After 21 days of culture with the other cells, enteroendocrine-differentiated NCI-H716 cells showed neuropods at their basolateral side and expressed their specific genes encoding proglucagon (*GCG*) and chromogranin A (*CHGA*). We showed that this model could be stimulated by commensal bacteria playing a key role in health, *Roseburia intestinalis* and *Bacteroides fragilis*, but also by a pathogenic strain such as *Salmonella Heidelberg*. Moreover, using cell-free supernatants of *B. fragilis* and *R. intestinalis*, we have shown that *R. intestinalis* supernatant induced a significant increase in *IL-8* and *PYY* but not in *GCG* gene expression, while *B. fragilis* had no impact. Our data indicated that *R. intestinalis* produced short chain fatty acids (SCFAs) such as butyrate whereas *B. fragilis* produced more propionate. However, these SCFAs were probably not the only metabolites implicated in *PYY* expression since butyrate alone had no effect. In conclusion, our new quadricellular model of gut epithelium could be an effective tool to highlight potential beneficial effects of bacteria or their metabolites, in order to develop new classes of probiotics.

## 1. Introduction

The intestinal microbiota contributes to the human health by maintaining, among others, the intestinal physiology [1]. A number of factors can disturb or alter the intestinal microbiota resulting in intestinal dysbiosis [2,3]. Emerging evidence in both animal and human models suggest that dysbiosis in gut microbiota may contribute to chronic diseases such as obesity and type 2 diabetes (T2D), which have become an increasing public health concern [2,4,5]. Abnormal changes in the gut microbiota cause intestinal epithelial cell damage and destroy the integrity of the intestinal barrier, which drives leakage of bacterial endotoxins and triggers systemic inflammation, thereby disrupting glucose metabolism [6]. However, gut microbiota studies in patients with T2D are limited and inconclusive. Further studies are required to increase the understanding of the complex interplay between intestinal microbiota and the host with T2D. To better understand how internal and external factors affect the commensal bacteria, in vitro models are needed. Most of the intestinal in vitro models are composed of human intestinal cell line Caco-2. However, these models do not reproduce the diversity of cell types present in the primary tissue, where several cells can be found such as Goblet, Paneth, and enteroendocrine cells. To overcome this limit, tricellular models where CaCo-2 was co-cultured with the mucus producing HT29-MTX cell line and M cells were developed and used to characterize gut bacteria such as *Bacteroides fragilis* [7,8].

The results of various studies showed that *Bacteroides* are decreased in T2D, as *Bifidobacterium*, *Faecalibacterium*, *Akkermansia,* and *Roseburia* [9]. *Bacteroides* spp., which constitute up to 30% of colonic microbiota, produce a variety of SCFAs: *B. fragilis* for instance, which is decreased in patients with T2D, produces acetate, butyrate, and mainly propionate [10,11]. *Roseburia* spp. belong to the *Firmicutes* phylum, including most human-gut associated butyrate producers [12,13]. Interest in *Roseburia* spp. has increased with reports showing that the abundance of these bacteria is also reduced in individuals affected by inflammatory diseases and colorectal cancers [13,14,15,16]. *R. intestinalis* plays an important role in the maturation of the immune system, primarily through the production of butyrate [13]. Some studies reported that butyrate could both increase and decrease the expression of Interleukin-8 (IL-8), which induces the recruitment of neutrophils to the infection site [17,18,19]. Intestinal SCFA concentrations are decreased in the presence of pathogenic bacteria such as those from *Enterobacteriaceae* family including *Salmonella* spp., which are often increased in T2D [20].

Increasing evidence suggests that SCFAs may play an important role in the pathogenesis of T2D. SCFAs appear to have a beneficial influence on glucose metabolism by normalizing blood glucose levels. Butyrate has received particular attention for its beneficial effects on both cellular energy metabolism and intestinal homeostasis [17,21,22]. Butyrate and other SCFAs (such as acetate and propionate) are therefore considered as pivotal endogenous signaling molecules, acting through their free fatty-acid receptors (FFAR) 2 and 3 [18]. Within the gut, the expression of these receptors has been localized to EECs, in particular, L-cells (L-ECC) [18]. SCFAs, by binding to L-ECC FFAR2 and FFAR3, can trigger the secretion of glucagon-like peptide-1 (GLP-1) and peptide YY (PYY) [10]. GLP-1 is an incretin hormone known to potentiate insulin secretion by the pancreas [11] and PYY induces satiety [23,24]. GLP-1 derives from proglucagon gene (*GCG*), expressed in both pancreatic cells and L-ECC [25]. Both GLP-1 and PYY can stimulate satiation, inhibiting ingestion, and attenuating glucose hyperglycemia acting either directly on the hypothalamic centers of appetite control, or indirectly via the vagal–brainstem–hypothalamic pathway [26]. Several studies indicated that commensal and pathogenic bacteria modulate gut hormone expression [20,27,28]. *Salmonella* increased hyperglycemia by decreasing GLP-1 secretion and increasing L-ECC apoptosis in *Salmonella*-infected piglets [20]. It was shown that that *Lachnoclostridium* spp. and *Butyricicoccus spp.* were positively correlated with an increase of GLP-1 production whereas *Prevotella copri* and *Ruminococcaceae* spp. are associated with insulin resistance [27]. *Lactobacillus* strains can stimulate GLP-1 production both in vitro and in vivo [28]. Altogether, these findings demonstrate the importance the bacterial metabolites SCFAs in the maintenance of metabolic health.

The aim of the current study was to investigate how two key commensal bacteria, *R. intestinalis* and *B. fragilis*, can impact secretion of GLP-1 and PYY in a new in vitro model of intestinal epithelium recapitulating the four main intestinal cell types (enterocytes, goblet cells, M cells, and enteroendocrine cells).

## 2. Materials and Methods

### 2.1. Cell Lines and Growth Culture

#### 2.1.1. Endocrine Cells (NCI-H716) Differentiation

Human NCI-H716 cells line, derived from a poorly differentiated adenocarcinoma of human cecum [29], was purchased from ATCC. Cells were maintained and grown in suspension in Roswell Park Memorial Institute medium (RPMI-1640) supplemented with 10% of fetal bovine serum (FBS), 1% of L-glutamine, 1% of 100 UI/mL penicillin, and 100 μg/mL streptomycin in a humidified incubator at 37 °C, 5% of CO₂. Cell adhesion and differentiation into endocrine cells were initiated by growing cells for 48 h in 24-well culture plate coated with Matrigel at 25 µg/cm² (Corning^®^) [8,30]. Caco-2 cells, HT29-MTX, and RajiB were cultivated as we described in Vernay et al. [8]. To evaluate the differentiation of NCI-H716, cells were stimulated in presence of glucose 1 g/L in phosphate buffer during 24 h.

#### 2.1.2. Cellular Model Composed of Four Cell Types (Quadricellular Model)

Human NCI-H716 cells were added to apical chamber of polycarbonate Transwell^®^ inserts containing 25 µg/cm² of Matrigel and incubated at 37 °C for 48 h to obtain endocrine differentiated NCI-H716 cells. Upon confluence, Caco-2 and HT29-MTX cells were harvested with trypsin–EDTA and a predetermined number of cells of each type were mixed prior to seeding on this insert containing endocrine differentiated NCI-H716 cells at a ratio of 8:1:1 (Caco-2:HT29-MTX: differentiated NCI-H716). After 14 days of culture, Raji B cells were added to the basolateral chamber to induce the differentiation of Caco-2 cells into M cells [8]. Caco-2/HT29-MTX/endocrine differentiated NCI-H716 cells and Raji B co-culture were maintained for 7 days in DMEM.

### 2.2. TEER Measurements

The integrity of the polarized epithelial co-culture (Caco-2: HT29-MTX: M: endocrine differentiated NCI-H716 cells) was evaluated as we previously described in Vernay et al. (2020) [8]. For the study of the impact of bacteria cell-free supernatant on the quadricellular model, transwells were transferred into cellZscope cell modules and TEER was measured hourly for 24 h using a cellZscope controller and software as described by [31].

### 2.3. Transmission Electron Microscopy

Transmission electron microscopy (TEM) was performed on quadricellular model cells after 21 days of growth on polycarbonate Transwell^®^ cell culture inserts as described in Vernay et al. (2020) [8]. Briefly, polarized cells were washed with PBS and were fixed for 2 h in room temperature in 2.5% glutaraldehyde dissolved in 0.1 M cacodylate. Samples were postfixed in 1% osmium tetroxide for 1 h at room temperature, rinsed in cacodylate buffer, and dehydrated in an ascending series of ethanol (70, 90, and 100%). The polycarbonate membrane on which the cells were grown was recovered and cut into thin strips. Samples were then infiltrated with an ascending concentration of Epon resin in ethanol mixtures. Finally, they were placed in fresh Epon for several hours and then embedded in Epon for 48 h at 60 °C. Resin blocks were sectioned into 80 nm ultrathin sections using LEICA UC7 ultramicrotome (LEICA Systems, Vienna, Austria). These sections were mounted on copper grids and stained. Grids were observed using a TEM JEOL-JEM 1400 (JEOL Ltd., Tokyo, Japan) at an accelerating voltage of 120 kV and equipped with a Gatan Inc. Orius 1000 camera.

### 2.4. Quantification of Selected Genes Expression Level by Quantitative Reverse Transcription PCR (RT-qPCR)

After 21 days of co-culture, RNAs were extracted from the upper chamber. RNAs were extracted using an RNA purification kit, including DNAse digestion on-column (MACHEREY-NAGEL), following manufacturer instructions. The purity of the RNAs was determined by analysis A260/A280 ratio using the NanoDrop spectrophotometer. RNAs were transcribed into cDNA with High-Capacity cDNA Reverse Transcription Kit (Applied Biosystems) and following the protocol provided by the manufacturer. The no reverse transcriptase controls were also set up along with sample reactions so as to confirm the absence of genomic DNA contamination. The quantitative RT-PCR amplification of cDNA was performed in a reaction mixture (10 μL) containing SYBR Green master mix and 1 μM primer pairs on a QuantStudioTM (Applied Biosystems) [32]. The primers used are listed in Table 1. Characteristic genes of each cell type were selected as [8]: sucrase isomaltase (SI), which is specific of Caco-2, mucin-2 (*MUC2*) secreted by HT29-MTX, metalloprotease-15 (*MMP15*) associated to M cells, and proglucagon (*GCG*), chromogranin A (*CHGA*), and peptide YY (*PYY*) for differentiated NCI-H716. Zona occludens-1 (*ZO-1*) and occludin (*OCLN*) were also studied to evaluate the presence of cells junctions. Interleukin-8 (*IL-8*) and transferrin receptor (*TFRC*) were investigated as they can be activated in the presence of bacteria inducing inflammation. Thermal cycling conditions included 95 °C for 15 s, 40 cycles of 95 °C for 3 s, and 60 °C for 1 min. Threshold cycle (Ct) values were determined with QuantStudioTM Real-Time qPCR v1.3 software. Ct values for each gene were normalized to housekeeping gene Ct values: TBP (TATA box binding protein). Fold-change values were calculated after normalization of each gene to the TBP internal control, using the comparative threshold method (Livak and Schmittgen, 2001), with quadricellular model without bacteria or supernatant as the reference condition. Results correspond to relative expression values, reported as the ratio of cells with bacteria/supernatant to cells without bacteria/supernatant.

### 2.5. Cytotoxicity Evaluation by Measuring LDH (Lactate Deshydrogenase) Release

Measure of lactate dehydrogenase (LDH) released was performed after incubating the quadricellular model with *S.* Heidelberg, *B. fragilis*, or *R. intestinalis* during 3 h. LDH was evaluated into the culture media using a Cytotoxicity Detection Kit (Roche) according to the manufacturer’s protocol. Results were normalized to a negative control (cells incubated with culture medium DMEM).

### 2.6. Bacteria Growth Conditions

*R. intestinalis* DSMZ-14610 was isolated on Columbia agar plates supplemented with 5% (*v*/*v*) defibrinated horse blood (AES Chemunex, Combourg, France), 25 mg/L of hemin, and 10 mg/L of menadione. *B. fragilis* ATCC 25285 was cultured as we have previously described in Vernay et al. (2020) [8]. The cultures were incubated during 48 h at 37 °C in an anaerobic chamber Macs-VA500 (Don Whitley) flooded with 80% N_2_, 10% H_2_, and 10% CO_2_. Colonies have been added in the Brain Heart Infusion (BHI) medium. In parallel, colonies were then added in complete DMEM medium containing 20% FBS, 1% L-glutamine. The last medium studied is the complete DMEM/BHI medium (1:1, *v*/*v*), which is an intermediate medium comprising a cell and bacterial culture medium. Following 24 h of incubation in an anaerobic chamber, culture media were collected and CFU enumeration was performed. To separate supernatant and bacterial pellets, the media were centrifuged at 3000× *g* for 10 min. Supernatant samples were sterilized in 0.22 µm filter and stored at −20 °C.

Fresh *Salmonella* Heidelberg were grown overnight at 37 °C in fresh Luria-Bertani (LB) as previously described in Vernay et al. (2020) [8].

### 2.7. Determination of Short Chain Fatty Acids (SCFAs) by Gas Chromatography Mass Spectrometry (GC/MS)

SCFA analysis in bacteria supernatants was performed by gas chromatography coupled to mass spectrometry (GC/MS). *R. intestinalis* and *B. fragilis* DMEM cultured were thawed and centrifuged at 18.000 g for 10 min. *R. intestinalis* and *B. fragilis* supernatants (100 μL) were combined with 20 μL of a mixture containing 10 µL of acetonitrile (5%) and 10 µL of methanol (80%). Then, 5 µL of 2 mg/mL ethyl-butyric acid (Sigma) used as an internal standard were added. The mixture was centrifuged at 5480 g during 10 min, and 100 µL was filtrated through 0.45-µm filters. A system composed of a 7820A gas chromatograph (Agilent Technologies Inc., Santa Clara, CA, USA) connected to a mass spectrometry (MS) 5975C detector (Agilent) was used to quantify the SCFAs. Data were collected with MassHunter GC/MS acquisition 10.1 software (Agilent Technologies, Inc. Santa Clara, CA, USA). Two microliters were directly injected into the gas chromatograph equipped with a ZB-WAX capillary column (30 m length by 0.25 mm internal diameter, with a 0.25 µm film thickness; Agilent) using H_2_ as the gas carrier, with a constant flow rate of 1.5 mL/min. The temperature of the injector was kept at 220 °C, and the injection was performed in splitless mode. Chromatographic conditions were as follows: an initial oven temperature of 50 °C, 2 min at 50 °C, 5 °C/min up to 120 °C, 2 min at 120 °C, 5 °C/min up to 180 °C, 5 min at 180 °C, and 5 °C/min up to 220 °C, 5 min at 220 °C. The column was directly connected to the MS detector, and the electron impact energy was set to 70 eV. The MS quadrupole was programmed at 150 °C and MS ion source at 230 °C. The data collected were in the range of 25 to 250 atomic mass units (at 3.18 scans/s). SCFAs were identified by comparing their mass spectra with those held in the NSIT Library database (NIST 17 Version 2.3) and by comparing their retention times with those of the corresponding standards (Volatile Free fatty acid mixture CRM46975) purchased from Sigma. Quantification was performed based on relative peak area of each SCFA adjusting the quantity of each compound based on that of the internal standard (IS). The quantity of each compound (µg) was equal to the ratio of relative peak area of each SCFA/relative peak area of IS * quantity of IS (µg).

### 2.8. Co-Incubation of the Quadricellular Model with Bacteria or Their Cell-Free Supernatant

Before incubation with the quadricellular model, *R. intestinalis* and *B. fragilis* were cultured in complete DMEM medium in anaerobic chamber during 24 h. *S.* Heidelberg was cultured in LB medium followed by incubation for 90 min at 37 °C [33]. This culture was then centrifuged at 3000× *g* for 15 min and the pellet was resuspended in complete DMEM, before addition to the apical side of the quadricellular model. In some experiments, after 21 days of coculture, the quadricellular model was incubated with pathogenic bacteria (*S.* Heidelberg) or with commensal bacteria (*B. fragilis* or *R. intestinalis*) at a multiplicity of infection (MOI) of 100, in DMEM without antibiotics. To avoid cell death induced by *S.* Heidelberg, bacteria and quadricellular model were incubated for 3 h only as described in [33], and epithelium integrity (TEER) and cytotoxicity (LDH) were measured. The cells were also lysed for gene expression analysis.

The second experiments were performed by an incubation of the apical side of the multicellular model with *R. intestinalis* or *B. fragilis* cell-free supernatants diluted two-fold in DMEM medium without antibiotics during 24 h, this dilution was used to avoid cell death. The cells were also stimulated by butyrate sodium (Sigma Aldrich, Saint-Quentin-Fallavier, France) at 1.26 mmol/L and propionate sodium (Sigma Aldrich) at 0.20 mmol/L. These concentrations corresponded to the average concentration of SCFAs found in *R. intestinalis* and *B. fragilis* two-fold diluted cell-free supernatant. After 24 h of incubation, cells were lysed to extract RNA for qPCR.

### 2.9. Statistical Analysis

All experiments were performed in biological and technical triplicates at least. After testing normality of data, distribution was confirmed using the Shapiro–Wilk test. If the distribution of the data was normal, the one-way ANOVA test was used otherwise the Kruskal–Wallis test was used. Data are presented as mean ± SEM and *p*-value less than 0.05 was considered as significant. The analysis of data and generation of the graph were performed by Graph Pad Prism software (Version 8.0).

## 3. Results

### 3.1. Differentiated NCI-H716 Cells Expressed Proglucagon (GCG) and Chromogranin A (CHGA) Genes

The NCI-H716 cells were differentiated into endocrine cells for 48 h using Matrigel at 25 µg/cm^2^. Differentiated NCI-H716 cells had morphological characteristics different from undifferentiated cells. NCI-H716 cells in vitro grew in suspension as floating aggregates, they were round in shape (Figure 1A). Matrigel-adherent differentiated NCI-H716 cells were filiform with larger cell body volume (Figure 1B). In addition, the differentiated cells had cytoplasmic extensions (black arrows), which may correspond to neuropods (Figure 1B) [32]. In the presence of Matrigel, the number of adherent (coated) cells are higher than untreated NCI-H716 (30.7% and 0.3%, respectively) (Figure 1C).

The effect of glucose on *GCG* expression, characteristic of intestinal endocrine L cells, was explored in endocrine-differentiated NCI-H716 cells. Undifferentiated NCI-H716 and differentiated NCI-H716 were stimulated in presence of glucose (1 g/L). A significant increase by 2.1-fold in *GCG* expression was detected between differentiated and undifferentiated NCI-H716 (Figure 1D). *CHGA* expression was also significantly increased by 1.4-fold, whereas no variation was observed for *PYY* expression after differentiation (Figure 1D).

### 3.2. Endocrine-Differentiated NCI-H716 Are Functional in Co-Culture with Caco-2, HT-29 MTX and M Cells

Caco-2, HT29-MTX, and endocrine-differentiated NCH-H716 cells were detected by optical microscopy after 5 or 7 days of coculture (Figure 2A,B, respectively). Endocrine-differentiated NCI-H716 cells established numerous cytoplasmic extensions as shown by black arrows. After 21 days of culture, all the cells were also present as indicated by electron transmission microscopy analysis. Figure 2C showed microvilli at the apical surface of endocrine-differentiated NCI-H716 and visible neuropods (as indicated by arrow) in the basolateral side. Moreover, these cells connected to Caco-2 displayed electron-dense cytoplasm typical of secretion vesicles (Figure 2C).

To confirm the presence of all cells after 21 days of culture, we analyzed the expression of genes specific of each cell type (Figure 2D). When compared to their expression in a tricellular model (with only Caco-2, HT29-MTX, and M cells), a significant increase by 710-fold was observed for *GCG* and a 13.53-fold increase for *CHGA*, indicating that in the quadricellular model, endocrine-differentiated NCI-H716 expressed their specific genes. The expression of *MUC2* (HT29-MTX), *SI* (Caco-2), and *MMP15* (M cells) was also detected but the difference was not significant compared to tricellular model (Figure 2D), showing that adding differentiated NCI-H716 did not modify the specificity of other cells.

We have measured the transepithelial electrical resistance (TEER) at 21 days of coculture to investigate epithelium barrier integrity (Figure 2E). When endocrine-differentiated NCI-H716 were cultured alone, TEER value was the same as that on insert without cells indicating that endocrine-differentiated NCI-H716 cells did not establish tight junctions between them. However, when endocrine-differentiated NCI-H716 cells were cocultured with other cells (Caco-2, HT29-MTX, and M cells), TEER was the same as in tricellular model containing only Caco-2, HT29-MTX, and M cells. Their TEER values were significantly higher than endocrine-differentiated NCI-H716 alone, showing that the addition of differentiated NCI-H716 did not affect tight junctions in quadricellular model. The quadricellular model including differentiated NCI-H716 cells was, therefore, well functional.

### 3.3. The Impact of Commensal and Pathogenic Bacteria on Quadricellular Model

We have investigated if *R. intestinalis* can grow in complete DMEM medium, in which the different cell types used for the quadricellular model were cultured. We compared complete DMEM to BHI, which is a specific bacterial culture medium. Moreover, a mixture of complete DMEM/BHI (1:1, *v*/*v*) was studied as an intermediate medium (Figure 3A,B). After 24 h of incubation, *R. intestinalis* showed a significantly greater growth in complete DMEM medium than BHI (an 8.94-fold increase). There was no significant difference in optical density at 600_nm_ (OD 600_nm_) between complete DMEM/BHI and complete DMEM (Figure 3A). These results were confirmed by enumeration of *R. intestinalis*, which was significantly greater in complete DMEM or in complete DMEM/BHI than in BHI (Figure 3B). Moreover, there was also no difference between complete DMEM and complete DMEM/BHI medium (Figure 3B). However, in order to test the effect of bacteria on quadricellular model, complete DMEM medium (cell culture medium) was used in this study. We have previously showed that *B. fragilis* cultivated in DMEM and can be used to characterize its interaction with intestinal epithelium [8].

Cultures of *R. intestinalis*, *B. fragilis,* or *S.* Heidelberg in complete DMEM were added to apical side of the quadricellular model. The co-culture for 3 h of the quadricellular model of intestinal epithelium with *R. intestinalis, B. fragilis,* or *S.* Heidelberg did not induce cell mortality as shown by the level of LDH release by the cells, which was not modified whatever the bacteria compared to the control complete DMEM (Figure 3C). As microbial organisms are known to induce expression of cytokines, we have evaluated the expression of *IL-8*. After 3 h of incubation, quadricellular model in presence of *B. fragilis* showed a significantly decreased by 9.26-fold of *IL-8* expression compared to the incubation with *S*. Heidelberg (Figure 3D). *R. intestinalis* also showed a low *IL-8* level compared to *S.* Heidelberg (a 2.45-fold decrease) infection but higher than *B. fragilis* (a 3.77-fold increase). We have also evaluated the expression of transferrin receptor (*TFRC*) gene, described as activated in presence of inflammatory cytokines [34]. *TFRC* was not significantly modified in presence of these bacteria after 3 h of incubation (Figure 3C). The impact of these bacteria on endocrine function was also evaluated by investigating *GCG* and *CHGA* expression. *B. fragilis* only increased significantly *GCG* expression (a 2.52-fold increase) whereas no significant difference was observed with *CHGA* (Figure 3C).

### 3.4. R. intestinalis and B. fragilis Produced SCFAs in DMEM

As each bacterial strain had specific effects without inducing cell death, we investigated if their cell-free supernatants could have an impact.

To determine if *R. intestinalis* and *B. fragilis* growing in complete DMEM produced SCFAs, a GC/MS analysis was conducted to examine the presence of the SCFAs in their supernatant. For *R. intestinalis* supernatant, butyrate (average 2.47 mmol/L) and acetate (average 0.40 mmol/L) were detected after 24 h of culture with a significant difference for butyrate only compared to complete DMEM without bacteria (Figure 4). For *B. fragilis* supernatant, a significant increase can be observed in the concentrations of acetate (1.43 mmol/L) and propionate (0.40 mmol/L) compared to complete DMEM and *B. fragilis* supernatant, whereas butyrate (0.99 mmol/L) was significantly increased compared to complete DMEM but significantly less than in *R. intestinalis* supernatant.

*R. intestinalis* and *B. fragilis* supernatants were tested in the quadricellular model to determine if the effects observed with the bacteria could be imitated by the supernatants. The butyrate and synthetic propionate conditions were added to investigate if the mediated effects by supernatants could be related to SCFAs. In the presence of *R. intestinalis* supernatant, an increase of *PYY* and *IL-8* genes expression was significantly different compared to untreated cells (Figure 5A). However, for *GCG*, *SI,* and *MUC2* expression, no significant difference was observed. Moreover, contrary to *R. intestinalis*, *B. fragilis* supernatant did not significantly induce the expression *PYY* (an 8.8-fold increase) or *IL-8* (a 118.4-fold increase). Moreover, contrary to the *R. intestinalis* supernatant, synthetic butyrate (at the same concentrations as in the two-fold diluted *R. intestinalis* supernatant) did not significantly increase *PYY* secretion. For other genes implicated in tight junctions such as *OCLD* and *ZO-1*, a significant decrease was observed compared to untreated cells with *R. intestinalis* supernatant (0.73 ± 0.25 and 0.69 ± 0.16-fold, respectively). However, this decrease did not seem to modify TEER as Cellzscope analysis showed that *R. intestinalis* supernatant significantly improved TEER (Figure 5B). After 13 h of incubation with *R. intestinalis* supernatant, the increase of TEER is improved by two-fold compared to the untreated cells and the difference was significant whereas in the presence of *B. fragilis* supernatant, TEER was also increased but not significantly.

## 4. Discussion

Emerging evidence suggests that dysbiosis in gut microbiota may contribute to diabetes; however, gut microbiota studies in patients with T2D are limited and inconclusive. To better understand how internal and external factors affect the commensal bacteria and their interaction with host, we have generated for the first time a quadricellular model including enteroendocrine cells (NCI-H716). At first, we have plated NCI-H716 to differentiate them into an endocrine lineage [35]. Several studies have indicated that this strategy does not alter levels of GLP-1 in these cells [36,37]. In our study, NCI-H716 became adherent endocrine-differentiated cells and expressed *GCG* in the presence of glucose but not *PYY* as shown by Kuhre et al. (2016) [38]. As endocrine cells comprise accounts for only 1% of the intestinal epithelium [39], and to be similar to human intestinal barrier, we have added L cells (NCI-H716) to the main other cells such as enterocytes (CaCo-2), mucus cells (HT29-MTX) and M cells [8]. Ref. [40] suggested that the co-culture of Caco-2 and NCI-H716 may be more suitable for the study of glucose transport than the Caco-2 model. Indeed, glucose transport was faster in coculture of Caco-2/NCI-H716 than monoculture of Caco-2 [40]. Moreover, in vivo, L cells are integrated within the epithelial cell layer and are therefore differentially exposed to luminal and plasma constituents at their apical and basolateral membrane surfaces [41]. We have addressed this limit by culturing the four cell types on insert transwell to have basal and apical side. After 21 days of culture with other cells, endocrine-differentiated NCI-H716 expressed their specific genes as *GCG* and *CHGA* (Figure 2D). Moreover, the presence of the endocrine-differentiated NCI-H716 did not change the function Caco-2 and HT29-MTX cells as their specific gene expression can be detected (Figure 2E).

This quadricellular model can be stimulated by commensal and pathogenic bacteria (Figure 3D). As the key role of two bacteria, *R. intestinalis* and *Bacteroides*, found in lower frequency in T2D, has not been fully described, we have analyzed their impact on the endocrine function. It was described that gut microbiota in T2D modified gut hormone secretion [18]. T2D being characterized by an expansion of facultative anaerobic *Enterobacteriaceae*, we used *S.* Heidelberg as pathogenic bacteria in this study [42]. We have shown that in the presence of bacteria, the expression of *IL-8* was more important for *S.* Heidelberg than *R. intestinalis* or *B. fragilis (*Figure 3D). It is well known that *Salmonella* activated *IL-8* by its flagellin recognition by TLR5 [43,44]. Moreover, *R. intestinalis*-derived flagellin has been proved as an effective modulator of inflammatory gut responses [45]. *B. fragilis* was described as promoting mucosal immune development and reduced inflammation [46,47,48]. For endocrine function, we also have a difference between the three bacteria. Only *B. fragilis* showed a significant increase of *GCG* expression compared to *S.* Heidelberg and *R. intestinalis (*Figure 3D). These results were not surprising as it was described by [49] that *Bacteroides* regulate bile acid metabolism by converting taurochenodeoxycholic acid to lithocholic acid, which activates G protein-coupled bile acid receptor-1 (TGR5) to stimulate GLP-1 secretion from L cells [49]. *B. fragilis* that we used in this study produced bile acids after growing in cell culture medium containing fetal bovine serum [50,51]. Interestingly, it was also reported that C57BL/6J mice reconstituted with *Bacteroides acidifaciens* had increased GLP-1 [52]. As *R. intestinalis* and *B. fragilis* had beneficial effects, we have tested their cell-free supernatant on the quadricellular model. We have evaluated the impact of these supernatants on the integrity of the epithelium by analyzing the expression of two genes, *ZO-1* and *OCLD*, implicated in cell junctions which constitute the epithelium barrier. *R. intestinalis* supernatant significantly decreased the expression of *ZO-1* and *OCLD,* but at low level which did not disturb TEER measurement (Figure 5A,B). Indeed, in the presence of *R. intestinalis,* TEER increased by two-fold after an incubation of 13 h. Moreover, even *R. intestinalis* supernatant induced *IL-8* expression compared to control and compared to *B. fragilis* supernatant, which did not activate *IL-8* expression. Decrease of *ZO-1* and *OCLD* in the presence of *R. intestinalis* can be correlated to the increase of *IL-8* expression [53]. We can speculate when *Roseburia* and *Bacteroides* are decreased, their beneficial effects are also decreased, which can explain that in T2D there is significantly enhanced permeability in the gut resulting in inflammation.

Moreover, we identified that *R. intestinalis* supernatant induced a significant increase in *PYY* but not *GCG* (Figure 5A), while *B. fragilis* appeared to have no impact. We have identified that these two bacteria are different in their capacity in secreting SCFAs (Figure 4). Indeed, in our conditions, *R. intestinalis* produced more butyrate whereas *B. fragilis* secreted more propionate and acetate. These results agree with other studies describing *B. fragilis* and *R. intestinalis* SCFAs [13,21,22,54,55]. However, when this quadricellular model was stimulated with synthetic butyrate sodium and propionate sodium at the same concentration as in the diluted supernatant, *PYY* expression was not significantly different from the control (Figure 5A). Several studies in isolated cell cultures suggested that SCFAs are direct enhancers of GLP-1 and PYY secretion [56], and a few human studies find increased GLP-1 and/or *PYY* secretion in response to rectally administered SCFAs [56,57]. Some studies showed that luminal and especially vascular infusion of acetate and butyrate significantly increases colonic GLP-1 secretion, and to a minor extent also *PYY* secretion. Propionate neither affected GLP-1 nor *PYY* secretion whether administered luminally or vascularly [57]. The difference between all these studies can be explained presumably because the majority of these SCFAs either remain in the colonic lumen or are absorbed and metabolized by the colonocytes [57]. In our study, butyrate sodium and propionate sodium did not induce a significant increase; these molecules are probably absorbed by Caco-2 and HT29-MTX. Moreover, the concentration of butyrate produced by *R. intestinalis* in vitro is lower (1.26 mmol/L) than in human where the total concentrations of SCFAs is approximately 20 to 40 mmol/L in ileum (Acetate/Propionate/Butyrate ratio 3:1:1) (https://www.frontiersin.org/articles/10.3389/fimmu.2019.00277/full#T2, accessed on 13 October 2022). We can speculate that other molecules produced by *R. intestinalis* probably activate PYY secretion. Indeed, it wasshown also that PYY secretion can be modulated by LPS, flagelin, and CpG, which have been shown to increase CCK secretion in STC-1 cells in in vivo models [58]. Moreover, by analyzing the metabolic products of the gut microbiota and their effects on host metabolism, SCFAs, branched-chain amino acids, bile acids signaling, and gut permeability might be remarkably linked to initiation and aggravation of T2D [59,60]. Other studies have shown that a low level of PYY was found in obese patients where *Roseburia* was found to be decreased [21,61]. Administration of *R. intestinalis* could restore the circulating *PYY* level, induce satiety, and cause a decrease in food intake. One study showed that bacterial proteins have been linked to satiety signaling such as *Escherichia coli* caseinolytic protease B (ClpB) stimulating the secretion of PYY in primary cell cultures of rat intestinal mucosa at nanomolar concentrations [62]. However, new studies are needed to better understand the specific molecular mechanisms that mediate the beneficial effect of *Roseburia intestinalis*.

## 5. Conclusions

To conclude, we have demonstrated in this study that enteroendocrine cells can be incorporated to cells mimicking enterocytes (Caco-2), mucus cells (HT29-MTX), and M cells without modifying their functions. The quadricellular model described here provides an in vitro system that could be a good alternative to further explore the roles of gut microbiota in T2D, by identifying which microbes are important and how they contribute to crosstalk. Due to the potential effects of GLP-1 and PYY, selecting bacteria inducing the secretion of these two molecules can be considered as the key to the development of these new classes of therapeutics. However, our study has limitations as only one strain was used, and we did not consider the bacterial diversity found in gut microbiota. More investigations are needed with a diverse and complex microbial community, playing a central role in human health.

## Figures and Tables

**Figure 1 microorganisms-10-02263-f001:**
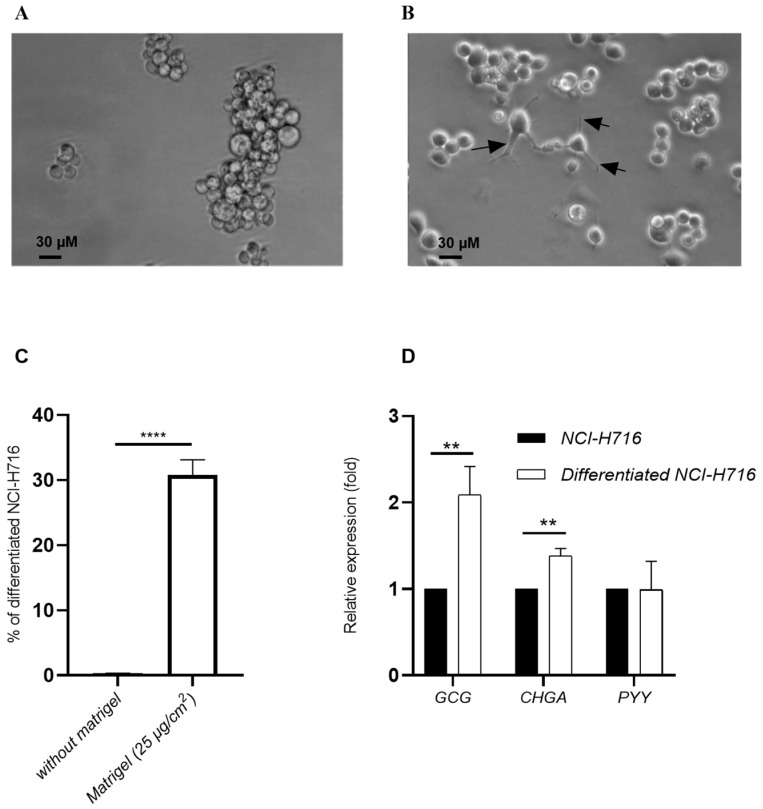
Differentiation of NCI-H716 to endocrine cells: Optical microscopy analysis of morphology of floating NCI-H716 cells (**A**) or coated endocrine-differentiated NCI-H716 cells with 25µg/cm^2^ of matrigel (**B**), arrows indicated neuropods. (**C**) Quantification of coated cells with neuropods after culturing NCI-H716 with matrigel. (**D**) NCI-H716 and differentiated NCI-H716 were stimulated with 1 g/L of Glucose and GCG, CHGA and PYY genes expression was evaluated. Normality of each condition was checked by Shapiro test and one-way ANOVA (in (**D**)) or Student test (in (**C**)) were used. ** *p* < 0.01; **** *p* < 0.0001.

**Figure 2 microorganisms-10-02263-f002:**
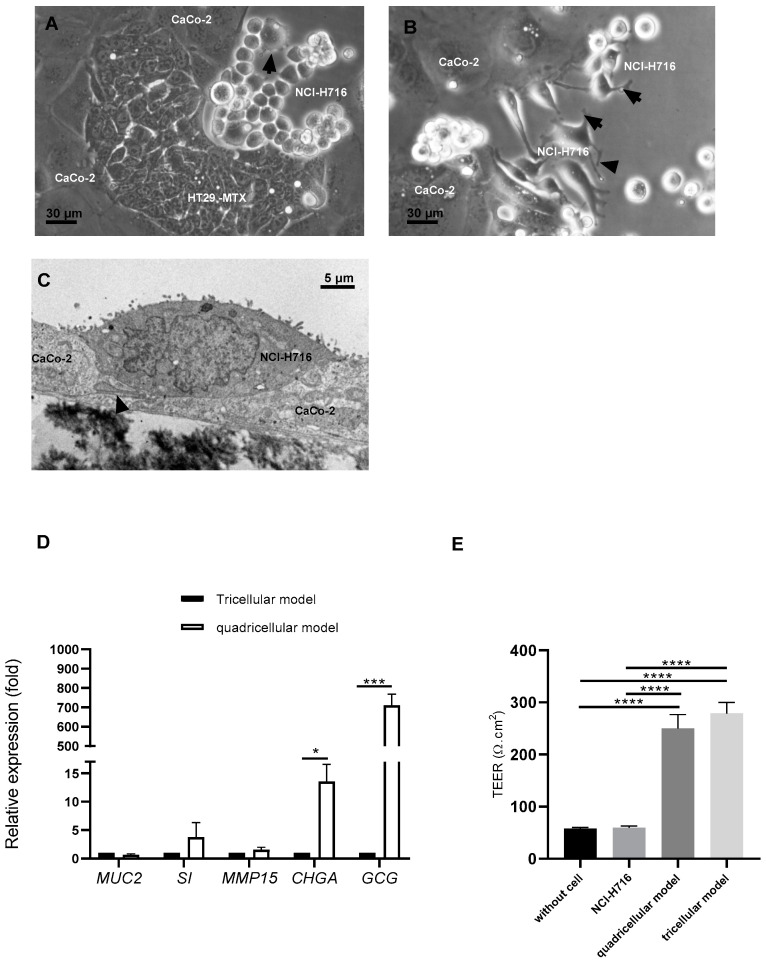
Endocrine-differentiated NCI-H716 in coculture with Caco-2/HT29-MTX and M cells (quadricellular model): Morphology analysis in optical microscopy ((**A**,**B**), bars represent 30 µm) or in transmission electron microscopy ((**C**), bar represents 5 µm), black arrows indicated neuropods, (**D**) analysis of relative specific genes expression compared to Caco-2/HT29-MTX/ M (tricellular model), and (**E**) evaluation of TEER at 21 days of culture for quadricellular and tricellular model. Normality of each condition was checked by Shapiro test and one-way ANOVA was used. * *p* < 0.05; *** *p* < 0.001; **** *p* < 0.0001.

**Figure 3 microorganisms-10-02263-f003:**
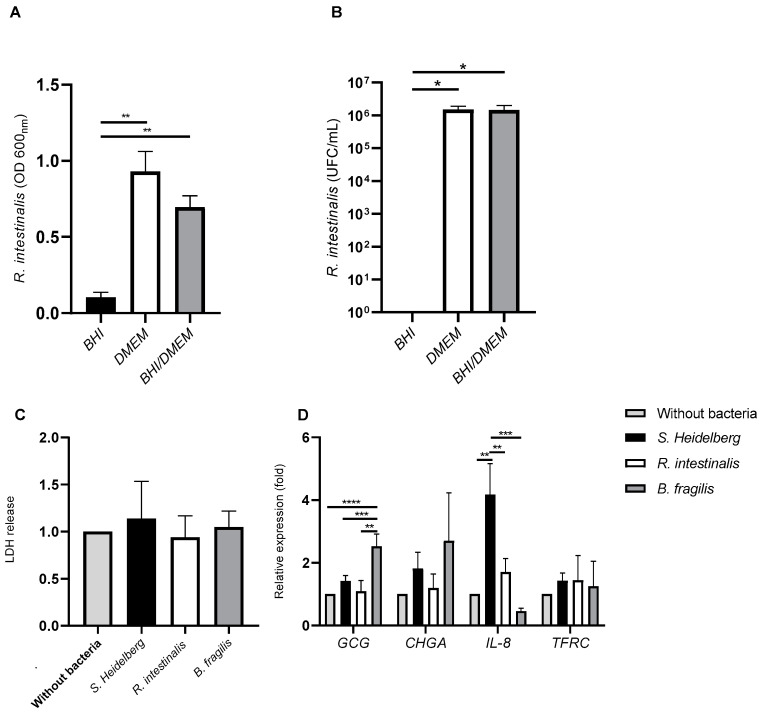
Impact of beneficial bacteria (*Roseburia intestinalis* and *Bacteroides fragilis*) and pathogenic bacteria (*S*. Heidelberg) on quadricellular model. (**A**,**B**) *R. intestinalis* and *B. fragilis* growth characterization, (**A**) overnight culture in BHI, or in DMEM media and in a medium containing a mixture of BHI and DMEM (1:1, *v*/*v*): (**B**) Enumeration of overnight culture in the same media. (**C**) LDH release after 3 h of incubation of bacteria with the quadricellular model. The results were normalized to cells without bacteria. (**D**) Analyze of genes expression implicated in endocrine (*GCG* and *CHGA*), in immune (*IL*-8) functions and transferrin receptor (*TFRC*) after 3 h of incubation. Normality of each condition was checked by Shapiro test and one-way ANOVA was used. * *p* < 0.05; ** *p* < 0.01; *** *p* < 0.001; and **** *p* < 0.0001.

**Figure 4 microorganisms-10-02263-f004:**
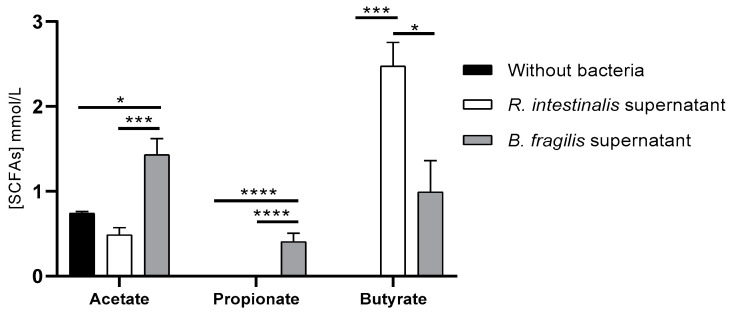
*R. intestinalis* and *B. fragilis* SCFAs production in complete DMEM: quantification of short fatty acids (acetate, propionate, and butyrate) in complete DMEM by GC-MS (mmol/L). Normality of each condition was checked by Shapiro test and one-way ANOVA was used. * *p* < 0.05; *** *p* < 0.001; **** *p* < 0.0001.

**Figure 5 microorganisms-10-02263-f005:**
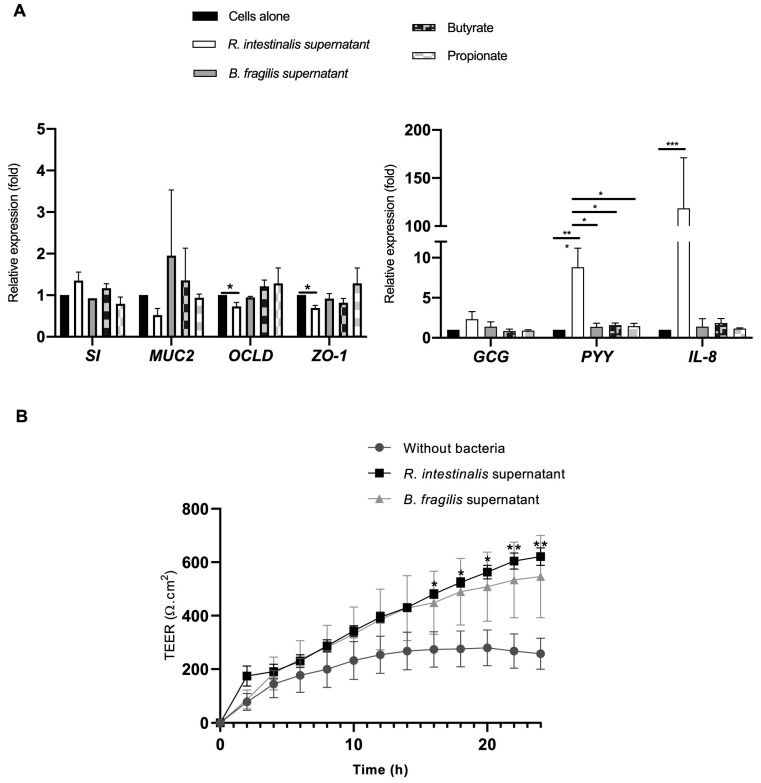
Stimulation of the quadricellular model with *R. intestinalis or B. fragilis* supernatants; with butyrate sodium (1.26 mmol/L) or propionate sodium (0.20 mmol/L): (**A**) Relative expression of specific genes implicated in barrier function (*SI*, *MUC2*, *OCLD,* and *ZO-1*), in endocrine function (*GCG* and *PYY*) and in immune response (*IL-8*) after 24 h of incubation; (**B**) impact on the TEER during 24 h for *R. intestinalis* and *B. fragilis* cell-free supernatant. Normality of each condition was checked by Shapiro test and Kruskal–Wallis test was used * *p* < 0.05; ** *p* < 0.01 and *** *p* < 0.001.

**Table 1 microorganisms-10-02263-t001:** Primers used in this study.

Gene	Encoded Protein	Left Primer	Right Primer
** * tbp * **	TATA Box protein	CCGGAATCCCTATCTTTAGTCC	GGGTCAGTCCAGTGCCATAAG
** *MUC2* **	Mucin- 2	CAGCACCGATTGCTGACTTG	GCTGGTCATCTCAATGGCAG
** *SI* **	Sucrase isomaltase	CATCCTACCATGTCAAGAGCCA	GCTTGTTAAGGTGGTCTGGTTT
** *MMP15* **	Metalloprotease 15	ACAACTATCCCATGCCCATC	ACCTGTCCTCTTGGAAGAAG
** *PYY* **	Peptide Tyrosine-Tyrosine	GCCTTGACCACAGTGCTTC	CTCTTTTCCCATACCGCTGC
** *GCG* **	Proglucagon	ACATTGCCAAACGTCACGA	GCGACCTCTTCTGGGAAATC
** *OCLN* **	Occludin	CCAATGTCGAGGAGTGGG	CGCTGCTGTAACGAGGCT
** *ZO-1* **	Zonula occludens 1	ATCCCTCAAGGAGCCATTC	CACTTGTTTTGCCAGGTTTTA
** *IL-8* **	Interleukin-8	AGACTTCCAAGCTGGCCGTGGCT	TCTCAGCCCTCTTCAAAAACTTCTC
** *CHGA* **	Chromogranin-A	CTACGCGCCTTGTCTCCTAC	AGTTGTGCCCAGTGGATAGG
** *TFRC* **	Transferrine receptor	GCTTTCCCTTTCCTTGCATATTCT	GGTGGTACCCAAATAAGGATAATCTGT

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
