# Peer review of "Roseburia intestinalis Modulates PYY Expression in a New a Multicellular Model including Enteroendocrine Cells"

_microorganisms, 2022, doi:10.3390/microorganisms10112263_

Round 1

Reviewer 1 Report

The purpose of the article “Roseburia intestinalis supernatant increases PYY expression in a multicellular model of intestinal epithelium including enteroendocrine cells in vitro”. The authors studied the to investigate how two key commensal bacteria, Roseburia intestinalis and Bacteroides fragilis, can impact secretion of GLP-1 and PYY in a new in vitro model of intestinal epithelium recapitulating the four main intestinal cell types. The article, however, must be improved in terms of writing since some grammar and syntax errors are present in the manuscript. They should address the subject and critically review the information from the literature.

 My suggestions:

1-     The authors need to revise the title of the paper in a more meaningful way.

2-     The abstract (introduction) is written in a way lacks logic. It should highlight the salient findings more critically.

3-     Keywords are present in the title, choose others.

4-     Introduction need more convincing rational for this article. 

5-     In item 2.3. Transmission Electron Microscopy (Material and Methods), detail the entire process of preparing the samples to be visualized and provide details on the procedures and protocols used during visualization.

6-     In item 2.4. Quatification of selected genes expression level by quatitative reverse transcription PCT (RT-qPCR) (Material and Methods), detailing all the chemical conditions of the PCR reactions, such as: RNA concentration, primer concentration .... as well as detailing the conditions physical factors inserted in the thermocycler, such as: temperature and annealing time of the primers and other conditions.

7-     As this is a 2-ddct relative expression analysis, describe the calibrator sample used to assess the relativity of expression levels.

8-     The results of this study are not fully explained therefore the interpretation of the relative expression results is very difficult. The author needs to provide the “x fold” increase or decrease rather than just writing ''significantly increased….''.

9-     Authors should discuss the results integrally. The discussion is based on individual results. I suggest that integrating the results will give more value to the work. The response of the plant to the drought and salinity is integrated. I suggest that you discuss by integrating all your results. You can use correlation tests (PCA or Pearson Correlation)

10-  In figures do the bars in the columns mean standard error or standard deviation? Improve caption description.

11-  Write the conclusion! It needs to be much improved.

Author Response

Thank you for your comments on our manuscript. We will reply to your questions one by one in the following paragraphs.

  • The authors need to revise the title of the paper in a more meaningful way.

We have changed the title “Roseburia intestinalis supernatant increases PYY expression in a multicellular model of intestinal epithelium including enteroendocrine cells in vitro” by “Roseburia intestinalis modulates PYY expression in a new a multicellular model including enteroendocrine cells”

  • The abstract (introduction) is written in a way lacks logic. It should highlight the salient findings more critically.

We have added for a better understanding “The gut microbiota contributes to human health and disease, however the mechanisms by which commensal bacteria interact with the host are still unclear. To date, a number of in vitro systems have been designed to investigate the host-microbe interactions. In the most of the intestinal model, the enteroendocrine cells, considered as a potential link between gut bacteria and several human diseases, were missing. In the present study, we have generated a new model by adding enteroendocrine cells (ECC) of L-type (NCI-H716) to the one that we have previously described including enterocytes, mucus and M cells. After 21 days of culture with the other cells, enteroendocrine-differentiated NCI-H716 cells showed neuropods at their basolateral side, and expressed their specific genes encoding proglucagon (GCG) and chromogranin A (CHGA). We showed that this model could be stimulated by commensal bacteria playing a key role in health, Roseburia intestinalis and Bacteroides fragilis, but also by pathogenic strain such as Salmonella Heidelberg.”

  • Keywords are present in the title, choose others.

We have changed the keywords : Hormone-producing cells; quadricellular model ; gut commensal bacteria; proglucagon; peptide tyrosine tyrosine

  • Introduction need more convincing rational for this article. 

We have changed the introduction by “A number of factors can disturb or alter the intestinal microbiota resulting in intestinal dysbiosis [2], [3]. Emerging evidences in both animal and human model suggest that dysbiosis in gut microbiota may contribute to chronic diseases such as obesity and type-2 diabetes (T2D), which have become an increasing public health concern [4–6]. Abnormal changes in the gut microbiota cause intestinal epithelial cell damage and destroy the integrity of the intestinal barrier, which drives leakage of bacterial endotoxins and triggers systemic inflammation, thereby disrupting glucose metabolism [7]. However gut microbiota studies in patients with T2D are limited and inconclusive. Further studies are required to increase the understanding of the complex interplay between intestinal microbiota and the host with T2D. To better understand how internal and external factors affect the commensal bacteria, in vitro models are needed. Most of intestinal in vitro models are composed of human intestinal cell line Caco-2. However, these models do not reproduce the diversity of cell types present in the primary tissue, where several cells can be found such as Goblet, Paneth and enteroendocrine cells. To overcome this limit, tricellular model where CaCo-2 were co-cultured with the mucus producing HT29-MTX cell line and M cells were developed and used to characterize gut bacteria such as Bacteroides fragilis [8,9].

The results of various studies showed that Bacteroides are decreased in T2D, as Bifidobacterium, Faecalibacterium, Akkermansia and Roseburia”.

 In item 2.3. Transmission Electron Microscopy (Material and Methods), detail the entire process of preparing the samples to be visualized and provide details on the procedures and protocols used during visualization.

We have modified this part by adding: “Briefly, polarized cells were washed with PBS and were fixed for 2 hours in room temperature in 2.5 % glutaraldehyde dissolved in 0.1 M cacodylate. Samples were postfixed in 1% osmium tetroxide for 1 hour at room temperature, rinsed in cacodylate buffer, and dehydrated in an ascending series of ethanol (70%, 90% and 100%). The polycarbonate membrane on which the cells were grown was recovered and cut into thin strips. Samples were then infiltrated with an ascending concentration of Epon resin in ethanol mixtures. Finally, they were placed in fresh Epon for several hours and then embedded in Epon for 48 hours at 60°C. Resin blocks were sectioned into 80 nm ultrathin sections using LEICA UC7 ultramicrotome (LEICA Systems, Vienna, Austria). These sections were mounted on copper grids and stained. Grids were observed using a TEM JEOL-JEM 1400 (JEOL Ltd, Tokyo, Japan) at an accelerating voltage of 120 kV and equipped with a Gatan Inc. Orius 1000 camera.

  • In item 2.4. Quatification of selected genes expression level by quatitative reverse transcription PCT (RT-qPCR) (Material and Methods), detailing all the chemical conditions of the PCR reactions, such as: RNA concentration, primer concentration .... as well as detailing the conditions physical factors inserted in the thermocycler, such as: temperature and annealing time of the primers and other conditions.

We have modified this part by “After 21 days of co-culture, RNAs were extracted from the upper chamber. RNAs were extracted using an RNA purification kit, including DNAse digestion on-column (MACHEREY-NAGEL), following manufacturer instructions. The purity of the RNAs was determined by analysis A260/A280 ratio using the NanoDrop spectrophotometer. RNAs were transcribed into cDNA with High Capacity cDNA Reverse Transcription Kit (Applied Biosystems) and, following the protocol provided by the manufacturer. The no reverse transcriptase controls were also set up along with sample reactions so as to confirm the absence of genomic DNA contamination. The quantitative RT-PCR amplification of cDNA, was performed in a reaction mixture (10 μL) containing SYBR Green master mix and 1 μM primer pairs on a QuantStudioTM (Applied Biosystems) [34]. The primers used are listed in table 1. Characteristic genes of each cell type were selected as described in Vernay et al. (2020): sucrase isomaltase (SI), which is specific of Caco-2, mucin-2 (MUC2) secreted by HT29-MTX, metalloprotease-15 (MMP15) associated to M cells, and proglucagon (GCG), chromogranin A (CHGA) and peptide YY (PYY) for differentiated NCI-H716. Zona occludens-1 (ZO-1) and occludin (OCLN) were also studied to evaluate the presence of cells junctions. Interleukin-8 (IL-8) and transferrin receptor (TFRC) were investigated as they can be activated in the presence of bacteria inducing inflammation. Thermal cycling conditions included 95 °C for 15 s, 40 cycles of 95 °C for 3 s, and 60 °C for 1min. Threshold cycle (Ct) values were determined with QuantStudioTM Real-Time qPCR v1.3 software. Ct values for each gene were normalized to housekeeping gene Ct values: TBP (TATA box binding protein). Fold-change values were calculated after normalization of each gene to the TBP internal control, using the comparative threshold method (Livak and Schmittgen, 2001), with quadricellular model without bacteria or supernatant as the reference condition in the case of the study of the impact of bacteria or their supernatant on quadricellular model. Results correspond to relative expression values, reported as the ratio of cells with bacteria/supernatant to cells without bacteria/supernatant.”

  • As this is a 2-ddctrelative expression analysis, describe the calibrator sample used to assess the relativity of expression levels.

We have added this for a better understanding: “Fold-change values were calculated after normalization of each gene to the TBP internal control, using the comparative threshold method (Livak and Schmittgen, 2001), with quadricellular model without bacteria or supernatant as the reference condition in the case of the study of the impact of bacteria or their supernatant on quadricellular model. Results correspond to relative expression values, reported as the ratio of cells with bacteria/supernatant to cells without bacteria/supernatant”

  • The results of this study are not fully explained therefore the interpretation of the relative expression results is very difficult. The author needs to provide the “x fold” increase or decrease rather than just writing ''significantly increased….''.

We have added in the all manuscript.

  • Authors should discuss the results integrally. The discussion is based on individual results. I suggest that integrating the results will give more value to the work. The response of the plant to the drought and salinity is integrated. I suggest that you discuss by integrating all your results. You can use correlation tests (PCA or Pearson Correlation). 

We have modified the discussion as follow: “Emerging evidence suggests that dysbiosis in gut microbiota may contribute to the diabetes, however gut microbiota studies in patients with T2D are limited and inconclusive. To better understand how internal and external factors affect the commensal bacteria and their interaction with host, we have generated for the first time a quadricellular model including enteroendocrine cells (NCI-H716). At first, we have plated NCI-H716 to differentiate them into an endocrine lineage [37]. Several studies have indicated that this strategy does not alter levels of GLP-1 in these cells [38,39]. In our study, NCI-H716 became adherent endocrine-differentiated cells and expressed GCG in the presence of glucose but not PYY as shown by Kuhre et al. (2016) [40]. As endocrine cells comprise accounts for only 1% of the intestinal epithelium [41], and to be similar to human intestinal barrier, we have added L cells (NCI-H716) to the main other cells such as enterocytes (CaCo-2), mucus cells (HT29-MTX) and M cells [9]. Xie et al. (2020) suggested that the co-culture of Caco-2 and NCI-H716 may be more suitable for the study of glucose transport than the Caco-2 model. Indeed, glucose transport was faster in coculture of Caco-2/NCI-H716 than monoculture of Caco-2 [42]. Moreover, in vivo, L cells are integrated within the epithelial cell layer and are therefore differentially exposed to luminal and plasma constituents at their apical and basolateral membrane surfaces [43]. We have addressed this limit by culturing the four cell types on insert transwell to have basal and apical side. After 21 days of culture with other cells, endocrine-differentiated NCI-H716 expressed their specific genes as GCG and CHGA (figure 2D). Besides, the presence of the endocrine-differentiated NCI-H716 did not change the function Caco-2 and HT29-MTX cells as their specific gene expression can be detected (figure 2E).

This quadricellular model can be stimulated by commensal and pathogenic bacteria (Figure 3D). As the key role of two bacteria, R. intestinalis and Bacteroides, found in lower frequency in T2D, has not been fully described, we have analyzed their impact on the endocrine function. It was described that gut microbiota in T2D modified gut hormone secretion [19]. T2D being characterized by an expansion of facultative anaerobic Enterobacteriaceae, we used S. Heidelberg as pathogenic bacteria in this study [45]. We have shown that in the presence of bacteria, the expression of IL-8 was more important for S. Heidelberg than R. intestinalis or B. fragilis (Figure 3D). It is well known that Salmonella activated IL-8 by its flagellin recognition by TLR5 [46,47]. Moreover, R. intestinalis derived flagellin has been proved as an effective modulator of inflammatory gut responses [48]. B. fragilis was described as promoting mucosal immune development and reduced inflammation [49–51]. For endocrine function, we have also a difference between the three bacteria. Only B. fragilis showed a significant increase of GCG expression compared to S. Heidelberg and R. intestinalis (Figure 3D). These results were not surprising as it was described by Pathak et al. (2018) that Bacteroides regulate bile acid metabolism by converting taurochenodeoxycholic acid to lithocholic acid, which activates G protein-coupled bile acid receptor-1 (TGR5) to stimulate GLP-1 secretion from L cells [52]. B. fragilis that we used in this study produced bile acids after growing in cell culture medium containing serum fetal bovine (Gautier et al. 2022 (doi: 10.3389/fmicb.2022.1023315) ; [53]). Interestingly, it was also reported that C57BL/6J mice reconstituted with Bacteroides acidifaciens had increased GLP-1 [54]. As R. intestinalis and B. fragilis had beneficial effect, we have tested their cell-free supernatant on the quadricellular model. We have evaluated the impact of these supernatant on the integrity of the epithelium by analyzing the expression of two genes, ZO-1 and OCLD, implicated in cell junctions which constitute epithelium barrier. R. intestinalis supernatant significantly decreased the expression of ZO-1 and OCLD, but at low level which did not disturb TEER measurement (figures 5A, 5B). Indeed, in the presence of R. intestinalis, TEER increased by two-fold after an incubation of 13h. Besides, even R. intestinalis supernatant induced IL-8 expression compared to control and compared to B. fragilis supernatant which did not activate IL-8 expression. Decrease of ZO-1 and OCLD in the presence of R. intestinalis can be correlated to the increase of IL-8 expression [55]. We can speculate when Roseburia and Bacteroides are decreased, their beneficial effect are also decrease which can explain that in T2D there is significantly enhanced permeability in the gut resulting in inflammation.

Besides, we identified that R. intestinalis supernatant induced a significant increase in PYY but not GCG (figure 5A), while B. fragilis appeared to have no impact. We have identified that these two bacteria are different in their capacity in secreting SCFAs (figure 4). Indeed, in our conditions, R. intestinalis produced more butyrate whereas B. fragilis secreted more propionate and acetate. These results are agreeing with other studies describing B. fragilis and R. intestinalis SCFAs [22,23,57–59]. However, when this quadricellular model was stimulated with synthetic butyrate sodium and propionate sodium at the same concentration as in the diluted supernatant, PYY expression was not significantly different from the control (figure 5A). Several studies in isolated cell cultures suggested that SCFAs are direct enhancers of GLP-1 and PYY secretion [60], and a few human studies find increased GLP-1 and/or PYY secretion in response to rectally administered SCFAs [60,61]. Some studies showed that luminal and especially vascular infusion of acetate and butyrate significantly increases colonic GLP-1 secretion, and to a minor extent also PYY secretion. Propionate neither affected GLP-1 nor PYY secretion whether administered luminally or vascularly [61]. The difference between all these studies can be explained by presumably because the majority of these SCFAs either remain in the colonic lumen or are absorbed and metabolized by the colonocytes [61]. In our study, butyrate sodium and propionate sodium did not induce significant increase, these molecules are probably absorbed by Caco-2 and HT29-MTX. Besides, the concentration of butyrate produced by R. intestinalis in vitro is lower (1.26 mmol/L) than in human where the total concentrations of SCFAs is approximately 20 to 40 mmol/L in ileum (Acetate/Propionate/Butyrate ratio 3:1:1) [68]. We can speculate that others molecules produced by R. intestinalis probably activate PYY secretion. Indeed, Covasa et al. (2019) showed that PYY secretion can be modulated by LPS, flagelin, and CpG, which have been shown to increase CCK secretion in STC-1 cells in vivo models [62] . Moreover, by analyzing the metabolic products of the gut microbiota and their effects on host metabolism, SCFAs, branched-chain amino acids, bile acids signaling, and gut permeability might be remarkably linked to initiation and aggravation of T2D [63,64]. Other studies have shown that a low level of PYY was found in obese patients where Roseburia was found decreased [22,65]. Administration of R. intestinalis could restore the circulating PYY level, induce satiety and a decrease in food intake. One study showed that bacterial proteins have been linked to satiety signaling such as Escherichia coli caseinolytic protease B (ClpB) stimulating the secretion of PYY in primary cell cultures of rat intestinal mucosa at nanomolar concentrations [66]. However, new studies are be needed to better understand the specific molecular mechanisms that mediate the beneficial effect of Roseburia intestinalis.”

In figures do the bars in the columns mean standard error or standard deviation? Improve caption description.

We have described it in line 242 “Statistical analysis: “Data are presented as mean ± SEM and p-value less than 0.05 was considered as significant”

  • Write the conclusion! It needs to be much improved.

We have modified the conclusion by adding:  

“5. Conclusion

To conclude, we have demonstrated in this study, that enteroendocrine cells can be incorporate to cells mimicking enterocytes (Caco-2), mucus cells (HT29-MTX) and M cells without modifying their functions. The quadricellular model, described here, provide an in vitro system that could be a good alternative to further explore the roles of gut microbiota in T2D, by identifying which microbes are important and how they contribute to crosstalk. Because of the potential effects of GLP-1 and PYY, selecting bacteria inducing the secretion of these two molecules can be considered as the key to the development of these new classes of therapeutics. However, our study has limitation as only one strain was used and did not consider the bacterial diversity found in gut microbiota. More investigations are needed with a diverse and complex microbial community, playing a central role in human health.”

Reviewer 2 Report

Major comments/questions:

1.      Statement in the abstract “differentiated NCI-H716 showed microvilli at the apical surface and visible neuropod in the basolateral side” is based on one microscopy image presented on figure 1C. Could you present any additional/quantitative data on the frequency of differentiation.

2.      images – „Light“ and „dark“ is quite misleading term – cell color on the image depends usually on the focal plane. In addition, scale bar on the images would be more appropriate than magnification (probably reflecting only magnification of the objective and not the total magnification).

3.      Fig 3A – it is confusing to the reader is 100% LDH release is actually positive control without cell death (and probably no LDH release). It would be better to have it normalized to the Triton X-100 control where all the cells are lysed to get the actual 100%. If such control was not done then I suggest to present data so that it is normalized to 1 (not %).

4.      Not clear from materials and methods how B. fragilis and R. intestinalis where precultured before adding to bacteria to quadricellular model.

5.      Reason of presenting data of unsuccessful culturing R. intestinalis in BHI is not clear (Figure 4 A and B). Why this data is important? There are many media in which bacteria are not growing. As DMEM worked fine, then I also can’t understand why 50:50 mixture of two media were used and not DMEM only?

6.      Figure 4C – does „without bacteria“ means BHI/DMEM media? If so please correct on the graph.

7.      Discussion – please reference figures in the manuscript also in discussion section – it is hard to follow which statements are based on current work and which on literature data.

8.      Discussion – please consider that PYY mRNA expression does not equal PYY secretion. I would consider the statement on line 430 over-interpretation.  I would also like to draw attention on Fig. 1c where PYY gene expression was not detected. I understand that the same cell type is expected to be responsible for  PYY gene expression in quadricellular model?

Minor comments:

1.      Typo in figure 2 legend „optimal microscopy“ – should be „optical“

2.      Line 358 “culturing the four cells”  - should be “four cell types”

3.      Figure 3. On A panel there is bacterium named S. Heidelberg and on panel B Salmonella – please harmonize.

Author Response

We feel greatly thankful for your professional reviewing on our article. According to your precious comments, we have revised the manuscript carefully. 

  1. Statement in the abstract “differentiated NCI-H716 showed microvilli at the apical surface and visible neuropod in the basolateral side” is based on one microscopy image presented on figure 1C. Could you present any additional/quantitative data on the frequency of differentiation.

- We have added a quantification of differentiated NCI-H716 without or with matrigel (25 µg/cm2) in figure 1C by enumerating the number of adherent cells with neuropods.

- We have added in lines 254-255: “In the presence of Matrigel, the number of adherent (coated) cells are higher than untreated NCI-H716 (30.7% and 0.3 %respectively) (Figure 1C).”

- In lines 267-268, we have added “(C) Quantification of adherent cells with neuropods after culturing NCI-H716 with Matrigel”.

  1. images – „Light“ and „dark“ is quite misleading term – cell color on the image depends usually on the focal plane. In addition, scale bar on the images would be more appropriate than magnification (probably reflecting only magnification of the objective and not the total magnification).

- We have deleted “Darker” and “light” terms.

- We have also added a scale for each image and captions in Figures 1A, B and in Figures 2A, B, C.

  1. Fig 3A – it is confusing to the reader is 100% LDH release is actually positive control without cell death (and probably no LDH release). It would be better to have it normalized to the Triton X-100 control where all the cells are lysed to get the actual 100%. If such control was not done then I suggest to present data so that it is normalized to 1 (not %).

Thank you for this comment, as we did not use Triton X-100 treatment, we have normalized the data to 1 in the figure 3C, as you suggested.

  1. Not clear from materials and methods how B. fragilis and R. intestinalis where precultured before adding to bacteria to quadricellular model.

- In lines 219-223, we added how bacteria are cultured before adding them on the apical side of the quadricellular model: “Before incubation with the quadricellular model, R. intestinalis and B. fragilis were cultured in complete DMEM medium in anaerobic chamber during 24h. S. Heidelberg was cultured in LB medium followed by incubation for 90 min at 37 °C [33]. This culture was then centrifuged at 3000 × g for 15 min and the pellet was resuspended in complete DMEM, before addition to the apical side of the quadricellular model.” 

- We have also indicated in lines 309-324“We have investigated if R. intestinalis can grow in complete DMEM medium, in which the different cells types used for the quadricellular model were cultured. We compared complete DMEM to BHI, which is a specific bacterial culture medium. Besides, a mixture of complete DMEM/BHI (1:1, v:v) was studied as an intermediate medium (Figures 4A, B). After 24h of incubation, R. intestinalis showed a significant greater growth in complete DMEM medium than BHI (an 8.94-fold increase). There was no significant difference in optical density at 600nm (OD 600nm) between complete DMEM/BHI and complete DMEM (Figure 4A). These results were confirmed by enumeration of R. intestinalis, which was significantly greater in complete DMEM or in complete DMEM/BHI than in BHI (figure 4B). Besides, there was also no difference between complete DMEM and complete DMEM/BHI medium (Figure 4B). However, in order to test the effect of bacteria on quadricellular model, complete DMEM medium (cell culture medium) was used in this study. We have previously showed that B. fragilis cultivated in DMEM and can be used to characterize its interaction with intestinal epithelium [35]. Cultures of R. intestinalis, B. fragilis or S. Heidelberg in complete DMEM were added to apical side of the quadricellular model.”

  1. 5.      Reason of presenting data of unsuccessful culturing R. intestinalis in BHI is not clear (Figure 4 A and B). Why this data is important? There are many media in which bacteria are not growing. As DMEM worked fine, then I also can’t understand why 50:50 mixture of two media were used and not DMEM only?

When we have started the project on Roseburia intestinalis, we did not know if it can growth in complete DMEM, so it why we have also used a mixture of DMEM and BHI and studied its growth in the different conditions.

For a better understanding of the study concerning the choice of the culture medium for R.intestinalis, we have added to Figure 3 the results concerning the growth in the different media (Figures 3A, B). The objective of this part was to be sure that R.intestinalis grew in DMEM (eukaryote cellular medium) compared to BHI (bacteria specific medium). As control, we used also a mix of BHI/DMEM if R. intestinalis did not grow in DMEM. The DMEM medium was chosen, as it was more appropriate in view of the incubation of bacteria with cells, as the BHI medium could be toxic to cells.

-We added in figure 3 caption: (A and B) R. intestinalis and B. fragilis growth characterization, (A) overnight culture in BHI, or in DMEM media and in a medium containing a mixture of BHI and DMEM (1:1, v:v): (B) Enumeration of overnight culture in the same media.”

-In lines 310-320, we added this “We have investigated if R. intestinalis can grow in complete DMEM medium, in which the different cells types used for the quadricellular model were cultured. We compared complete DMEM to BHI, which is a specific bacterial culture medium. Besides, a mixture of complete DMEM/BHI (1:1, v:v) was studied as an intermediate medium (Figures 4A, B). After 24h of incubation, R. intestinalis showed a significant greater growth in complete DMEM medium than BHI (an 8.94-fold increase). There was no significant difference in optical density at 600nm (OD 600nm) between complete DMEM/BHI and complete DMEM (Figure 4A). These results were confirmed by enumeration of R. intestinalis, which was significantly greater in complete DMEM or in complete DMEM/BHI than in BHI (figure 4B). Besides, there was also no difference between complete DMEM and complete DMEM/BHI medium (Figure 4B). However, in order to test the effect of bacteria on quadricellular model, complete DMEM medium (cell culture medium) was used in this study. We have previously showed that B. fragilis cultivated in DMEM and can be used to characterize its interaction with intestinal epithelium [35].”

  1. Figure 4C – does „without bacteria“ means BHI/DMEM media? If so please correct on the graph.

The condition "without bacteria" corresponds to the condition where the quadricellular model was incubated only with complete DMEM (with FBS and glutamine). We replace without bacteria by without bacteria (only with DMEM) for a better understanding.

  1. Discussion – please reference figures in the manuscript also in discussion section – it is hard to follow which statements are based on current work and which on literature data.

We have modified it in the discussion, in which we have cited the figures to facilitate understanding of this part; “Emerging evidence suggests that dysbiosis in gut microbiota may contribute to the diabetes, however gut microbiota studies in patients with T2D are limited and inconclusive. To better understand how internal and external factors affect the commensal bacteria and their interaction with host, we have generated for the first time a quadricellular model including enteroendocrine cells (NCI-H716). At first, we have plated NCI-H716 to differentiate them into an endocrine lineage [37]. Several studies have indicated that this strategy does not alter levels of GLP-1 in these cells [38,39]. In our study, NCI-H716 became adherent endocrine-differentiated cells and expressed GCG in the presence of glucose but not PYY as shown by Kuhre et al. (2016) [40]. As endocrine cells comprise accounts for only 1% of the intestinal epithelium [41], and to be similar to human intestinal barrier, we have added L cells (NCI-H716) to the main other cells such as enterocytes (CaCo-2), mucus cells (HT29-MTX) and M cells [9]. Xie et al. (2020) suggested that the co-culture of Caco-2 and NCI-H716 may be more suitable for the study of glucose transport than the Caco-2 model. Indeed, glucose transport was faster in coculture of Caco-2/NCI-H716 than monoculture of Caco-2 [42]. Moreover, in vivo, L cells are integrated within the epithelial cell layer and are therefore differentially exposed to luminal and plasma constituents at their apical and basolateral membrane surfaces [43]. We have addressed this limit by culturing the four cell types on insert transwell to have basal and apical side. After 21 days of culture with other cells, endocrine-differentiated NCI-H716 expressed their specific genes as GCG and CHGA (figure 2D). Besides, the presence of the endocrine-differentiated NCI-H716 did not change the function Caco-2 and HT29-MTX cells as their specific gene expression can be detected (figure 2E).

This quadricellular model can be stimulated by commensal and pathogenic bacteria (Figure 3D). As the key role of two bacteria, R. intestinalis and Bacteroides, found in lower frequency in T2D, has not been fully described, we have analyzed their impact on the endocrine function. It was described that gut microbiota in T2D modified gut hormone secretion [19]. T2D being characterized by an expansion of facultative anaerobic Enterobacteriaceae, we used S. Heidelberg as pathogenic bacteria in this study [45]. We have shown that in the presence of bacteria, the expression of IL-8 was more important for S. Heidelberg than R. intestinalis or B. fragilis (Figure 3D). It is well known that Salmonella activated IL-8 by its flagellin recognition by TLR5 [46,47]. Moreover, R. intestinalis derived flagellin has been proved as an effective modulator of inflammatory gut responses [48]. B. fragilis was described as promoting mucosal immune development and reduced inflammation [49–51]. For endocrine function, we have also a difference between the three bacteria. Only B. fragilis showed a significant increase of GCG expression compared to S. Heidelberg and R. intestinalis (Figure 3D). These results were not surprising as it was described by Pathak et al. (2018) that Bacteroides regulate bile acid metabolism by converting taurochenodeoxycholic acid to lithocholic acid, which activates G protein-coupled bile acid receptor-1 (TGR5) to stimulate GLP-1 secretion from L cells [52]. B. fragilis that we used in this study produced bile acids after growing in cell culture medium containing serum fetal bovine (Gautier et al. 2022 (doi: 10.3389/fmicb.2022.1023315) ; [53]). Interestingly, it was also reported that C57BL/6J mice reconstituted with Bacteroides acidifaciens had increased GLP-1 [54]. As R. intestinalis and B. fragilis had beneficial effect, we have tested their cell-free supernatant on the quadricellular model. We have evaluated the impact of these supernatant on the integrity of the epithelium by analyzing the expression of two genes, ZO-1 and OCLD, implicated in cell junctions which constitute epithelium barrier. R. intestinalis supernatant significantly decreased the expression of ZO-1 and OCLD, but at low level which did not disturb TEER measurement (figures 5A, 5B). Indeed, in the presence of R. intestinalis, TEER increased by two-fold after an incubation of 13h. Besides, even R. intestinalis supernatant induced IL-8 expression compared to control and compared to B. fragilis supernatant which did not activate IL-8 expression. Decrease of ZO-1 and OCLD in the presence of R. intestinalis can be correlated to the increase of IL-8 expression [55]. We can speculate when Roseburia and Bacteroides are decreased, their beneficial effect are also decrease which can explain that in T2D there is significantly enhanced permeability in the gut resulting in inflammation.

Besides, we identified that R. intestinalis supernatant induced a significant increase in PYY but not GCG (figure 5A), while B. fragilis appeared to have no impact. We have identified that these two bacteria are different in their capacity in secreting SCFAs (figure 4). Indeed, in our conditions, R. intestinalis produced more butyrate whereas B. fragilis secreted more propionate and acetate. These results are agreeing with other studies describing B. fragilis and R. intestinalis SCFAs [22,23,57–59]. However, when this quadricellular model was stimulated with synthetic butyrate sodium and propionate sodium at the same concentration as in the diluted supernatant, PYY expression was not significantly different from the control (figure 5A). Several studies in isolated cell cultures suggested that SCFAs are direct enhancers of GLP-1 and PYY secretion [60], and a few human studies find increased GLP-1 and/or PYY secretion in response to rectally administered SCFAs [60,61]. Some studies showed that luminal and especially vascular infusion of acetate and butyrate significantly increases colonic GLP-1 secretion, and to a minor extent also PYY secretion. Propionate neither affected GLP-1 nor PYY secretion whether administered luminally or vascularly [61]. The difference between all these studies can be explained by presumably because the majority of these SCFAs either remain in the colonic lumen or are absorbed and metabolized by the colonocytes [61]. In our study, butyrate sodium and propionate sodium did not induce significant increase, these molecules are probably absorbed by Caco-2 and HT29-MTX. Besides, the concentration of butyrate produced by R. intestinalis in vitro is lower (1.26 mmol/L) than in human where the total concentrations of SCFAs is approximately 20 to 40 mmol/L in ileum (Acetate/Propionate/Butyrate ratio 3:1:1) [68]. We can speculate that others molecules produced by R. intestinalis probably activate PYY secretion. Indeed, Covasa et al. (2019) showed that PYY secretion can be modulated by LPS, flagelin, and CpG, which have been shown to increase CCK secretion in STC-1 cells in vivo models [62] . Moreover, by analyzing the metabolic products of the gut microbiota and their effects on host metabolism, SCFAs, branched-chain amino acids, bile acids signaling, and gut permeability might be remarkably linked to initiation and aggravation of T2D [63,64]. Other studies have shown that a low level of PYY was found in obese patients where Roseburia was found decreased [22,65]. Administration of R. intestinalis could restore the circulating PYY level, induce satiety and a decrease in food intake. One study showed that bacterial proteins have been linked to satiety signaling such as Escherichia coli caseinolytic protease B (ClpB) stimulating the secretion of PYY in primary cell cultures of rat intestinal mucosa at nanomolar concentrations [66]. However, new studies are be needed to better understand the specific molecular mechanisms that mediate the beneficial effect of Roseburia intestinalis.”

  1. Discussion – please consider that PYY mRNA expression does not equal PYY secretion. I would consider the statement on line 430 over-interpretation.  I would also like to draw attention on Fig. 1c where PYY gene expression was not detected. I understand that the same cell type is expected to be responsible for  PYY gene expression in quadricellular model?

- We have replaced secretion by expression.

- Sorry, this is an error in the graph, there is indeed a low expression of PYY in the cells. However, after differentiation of NCI-H716, there is no difference in PYY expression observed between differentiated and not differentiated NCI-H716.

- In line 261, we have added: “whereas no variation was observed for PYY expression after differentiation (figure 1D)”.

Minor comments:

  1. Typo in figure 2 legend „optimal microscopy“ – should be „optical“

We have replaced optimal by optical

  1. Line 358 “culturing the four cells”  - should be “four cell types”

We have changed culturing the four cells by culturing the four cell types

  1. Figure 3. On A panel there is bacterium named S. Heidelberg and on panel B Salmonella – please harmonize.

We have homogenized « Salmonella Heidelberg or S. Heidelberg » and in figure 3 and in all the manuscript.

Round 2

Reviewer 1 Report

The authors addressed all my requirements, and justified reasonable all their answers. The manuscript is considerably improved and clearer. I suggest to be accepted in the present form.